# Ending Intimate Partner Violence (IPV) and Locating Men at Stake: An Ecological Approach

**DOI:** 10.3390/ijerph16091652

**Published:** 2019-05-12

**Authors:** Immacolata Di Napoli, Fortuna Procentese, Stefania Carnevale, Ciro Esposito, Caterina Arcidiacono

**Affiliations:** Department of Humanities Università degli Studi di Napoli Federico II, Naples 80133, Italy; immacolata.dinapoli@mail.com (I.D.N.); fortuna.procentese@unina.it (F.P.); steffy79@alice.it (S.C.); c.esposito91@libero.it (C.E.)

**Keywords:** intimate partner violence, perpetrators, child witnesses, ecological approach, domestic gender-based violence

## Abstract

Interventions for ending intimate partner violence (IPV) have not usually provided integrated approaches. Legal and social policies have the duty to protect, assist and empower women and to bring offenders to justice. Men have mainly been considered in their role as perpetrators to be subjected to judicial measures, while child witnesses of violence have not been viewed as a direct target for services. Currently, there is a need for an integrated and holistic theoretical and operational model to understand IPV as gender-based violence and to intervene with the goal of ending the fragmentation of existing measures. The EU project ViDaCS—Violent Dads in Child Shoes—which worked towards the deconstruction and reconstruction of violence’s effects on child witnesses, has given us the opportunity to collect the opinions of social workers and child witnesses regarding violence. Therefore, the article describes measures to deal with IPV, proposing functional connections among different services and specific preventative initiatives. Subsequently, this study will examine intimate partner violence and provide special consideration to interventions at the individual, relational, organizational and community levels. The final goal will be to present a short set of guidelines that take into account the four levels considered by operationalizing the aforementioned ecological principles.

## 1. Introduction

Domestic violence occurs predominantly between intimate partners, encompassing emotional, psychological, economic, physical and sexual forms of violence, abuse, control, threatening behaviour and coercion [1,2]. The Istanbul conference [2] addressed violence against women and domestic violence, clarifying in Article 3 that “domestic violence occurs within the family or domestic unit or between former or current spouses or partners, whether or not the perpetrator shares or has shared the same residence with the victim.” In the literature, when violence occurs among a couple in the context of their reciprocal interactions, it is called intimate partner violence (IPV).

Indeed, the literature and national statistics demonstrate that the prevalent form of domestic violence consists of violence on the part of men against women, and that this violence frequently occurs in front of children [3,4,5,6,7]. Some authors [8,9,10] provide evidence that women can also be perpetrators of violence; however, from an epidemiological perspective, nearly all cases of murder among partners are committed by men, and women are more likely to be victims. 

According to the WHO, “the most widely used model for understanding violence is the ecological model, which proposes that violence is a result of factors operating at four levels: individual, relationship, community and societal. Researchers have begun to examine evidence at these levels in different settings, to understand better the factors associated with variations in prevalence; however, there is still limited research on community and societal influences. Some risk factors are consistently identified across studies from many different countries, while others are context specific and vary among and within countries (e.g., between rural and urban settings). It is also important to note that, at the individual level, some factors are associated with perpetration, some with victimization, and some with both” [1] (p. 3). 

When examining the interactions among different actors (both perpetrators and victims, as well as child witnesses) the ecological model allows us to detect the hidden role of cultural, legal and organizational assets in buffering IPV. In fact, the prosecution of IPV perpetrators depends heavily on the potential impact of specific measures and resources related to social and legal services, their protective factors, preventative strategies, and the rules set forth by the existing legal framework. Furthermore, a “contextual analysis can be used to improve our understanding of IPV episodes, which can be utilized to inform future IPV prevention and treatment program development by helping victims to identify and effectively respond in situations where they may be at a heightened risk for experiencing IPV” [11] (p. 105). The Istanbul conference [2] recommended that participating countries establish services and measures to
protect women victims and bring perpetrators to justice. Over the years, however, making these recommendations a reality has proven to be complicated. Protective measures require the awareness of family members and neighbours, as well as the integrated interventions of police and health and educational personnel. Victim and perpetrator are categories used to define people after violent events; however, to prevent IPV, we need general preventative measures directed at all men and women, which are designed to balance power among family members, improve emotion regulation skills and help both victims and perpetrators cope with their anger. 

Therefore, even if the ecological model is a general reference for IPV prevention [1], it primarily provides a framework outlining its core principles. As a result, there is still a need to improve the knowledge of certain specific articulations of IPV, as well as of IPV-related services organizations. 

### IPV and Gender-Based Violence 

Our ecological model outlines the reciprocal interconnections of different variables acting at specific levels.

The most innovative dimension of the article is its subsequent goal to frame some specific recommendations for social and psychological interventions from the ecological perspective. Therefore, this article will operationalize the ecological principles into specific operational guidelines for the organization of services.

The gender-based perspective taken by this article stresses that men and women do not hold the same amount of power in regard to their dynamics as a couple and among the family as a whole. It is a fact that couple conflict can degenerate into violent actions against women, which sometimes leads to them being killed. Even the recently introduced term femicide has no masculine equivalent. It is no coincidence that violence can be elicited or aggravated when women assume subjective roles that depart from fixed and gendered role assignments. In fact, in these cases, perpetrators of violence frequently refer to fantasies of cruel mothers or narcissistic women. Moreover, when women fight against male supremacy or challenge male authority, men often view this as a significant offense that makes them intimately vulnerable and impotent, threating their core identity. Therefore, the advent of more balanced social power structures makes men unable to cope with a new society where male privilege is being broken down. 

Recent figures and statistics provide evidence of the distribution of IPV among the population, in addition to its prevalence among male partners [7]. However, only by focusing on the psychological dimensions of perpetrators can IPV be understood as solely an individual or relational pathology. Therefore, an ecological approach that is able to focus on the individual dimensions of a more general issue of social power is necessary.

As a result, this integrated and holistic theoretical and operational model for understanding and intervening against IPV will also overcome the fragmentation of services directed at IPV due to their different goals. This ecological approach also proposes a perspective that places value on interventions directed not only at victims, but also at perpetrators, in the context of both preventative and therapeutic perspectives. 

Moreover, this article examines the impact of IPV among couples on child witnesses. Since the 1970s, the phenomenon of witnessing domestic violence was considered to fall within the scope of psychological mistreatment [12,13,14] because it significantly disrupted children’s socialization [15], bringing about emotional, psychological, behavioural and social problems [16,17,18,19,20]. Although not all children who witness domestic violence develop these types of problems, they often develop conflict resolution methods characterized by the use of violence and, quite frequently, they believe that they are responsible for violence between their parents [17]. The consequences of continuous exposure to inter-parental violence also differ according to sex and age. Girls are more at risk in adolescence, while boys are more at risk during childhood. Children who witness violence also express behavioural and relational difficulties; therefore, they have recently been recognized as a specific target for treatment and preventative interventions.

## 2. A Theoretical and Operational Model for Ending IPV

The ecological perspective [21] is a holistic and multidimensional model for understanding and intervening to end IPV and its effects. It is a multidimensional approach that is widely supported to identify the risk factors for violent behaviour [22,23,24]. The ecological approach integrates different levels for taking action, including the collective/community, organizational, relational and individual levels. While these levels have previously been analysed in the literature, most analyses of them have examined their impacts separately. This ecological model will describe feminist, cognitive behavioural and psychodynamic approaches, following a sequence moving from the collective to the individual levels. 

Our study seeks to better understand the role of perpetrators of IPV and potential ways to deal with them. This is the defining characteristic of our approach, arising from the EU ViDaCS (Violent Dads in Child shoes) project (EU REC-AG-2017/REC-RDAP-GBV-AG-2017); grant 268650—of which some of the results referring to interventions specifically directed at perpetrators are reported below. 

Furthermore, the ecological model primarily addresses intervention against IPV by adopting a complexity paradigm and a systemic approach that also includes child witnesses in its framework. The challenge is to ensure the maximum dissemination of the proposed model among all of those involved in contrasting gender violence services with an emphasis on IPV, creating a productive and collaborative network that intercepts stories of invisible IPV reaching men who use violence against their partners and who are not involved in legal proceedings. Thus, our main goal is to permit the invisible actors, particularly socially integrated and “unsuspecting’” men, to deal with disruptive and intolerable emotional experiences, instead of defensively blaming their partners for their own violent behaviour. 

### 2.1. The Collective/Community Level

An ecological perspective assumes that the collective level in its cultural, legislative and political elements has a strong influence on the construction of a collective perception of gender violence and, consequently, a strong impact on individual psychological growth. Different approaches have focused on specific dimensions of this collective perception, but, at a social level, they mostly consider cultural dimensions expressed through social representations and the methods, by which social, as well as intimate, relationships are established.

The social approach observes the social context and considers how norms, stereotypes, gender roles, and attitudes towards violence influence relationships between men and women [25,26,27]. According to this approach, violence against women within a couple is legitimized in the context of adherence to traditional gender norms and roles. Specifically, a woman’s observance of female gender roles allows her to tolerate high levels of physical, sexual, psychological and emotional violence [28]. In contrast, men grow up surrounded by ideas of masculinity, culturally and socially supported by virility, which stress the importance of recognizing their power, strength and control. When men do not feel that these traits are being sufficiently recognized, they perceive their gender as being threatened, resulting in feelings of an identity gap. Acknowledging the disparity between men’s and women’s rights within the patriarchal culture is a potential first step towards raising awareness of the violence conveyed by gender stereotypes. This is the starting point for the deconstruction of “gender stereotypical models, relational forms between the sexes, and social models that ensure that women and men can be defined differently” [29] (p. 6).

The feminist perspective highlights the influence of patriarchal culture and its effects on women, men, family and society. In this approach, male partners perpetrate violence against women in the family in order to maintain control over them [30]. According to this perspective, IPV can result from the assumption that the male partner is automatically in a position of domination [31]. These beliefs regarding the social position of men are modelled on and culturally and socially acquired through their families of origin, their early childhood experiences, their life experiences and their social attitudes [11]. More specifically, gender stereotypes not only establish the characteristics designated as masculine and feminine, but, most importantly, outline the expectations related to male and female actions. From this point of view, violent episodes are induced by adherence to cultural rules and traditional gender roles that attribute to men a type of masculinity supported by virility that is recognized in power, strength and control over others. Women’s adherence to gender roles could be why they often present high tolerance to physical, sexual, psychological and emotional violence [28].

From the cognitive behavioural approach, the violent behaviour of perpetrators is also seen as being based on sexist attitudes, gender stereotypes and misogyny, all of which are typical of the patriarchal culture, and possess particular implications at the individual level. In fact, men learn specific and rigorous behaviours and scripts, imbued with gender stigma, that lead them to develop deficits in conflict management and in their ability to control their anger [32]. They develop a distorted understanding of relationships, hostile attitudes and beliefs, heightened executive functioning and a more spatial working memory [33]. According to the cognitive behavioural approach, interventions focused on perpetrators of IPV must be directed towards re-education, in order to increase their awareness of their behaviour and emphasise alternatives to violence [34].

The psychodynamic viewpoint sees violence against women in the context of the Freudian concept of the ‘rejection of femininity’ in both sexes [35]. Femininity, in itself, represents freedom, which rattles the primacy of the phallic identity and, therefore, triggers reactions of hate and sexist violence. In this vein, the patriarchal ideology has always tried to exorcise the ‘otherness’ of femininity in three ways. First, by affirming motherhood as the inescapable destiny of femininity, becoming a mother corresponds to the renunciation of being a woman, encapsulated by the idea that “the aberration of the patriarchal interpretation of motherhood is to think that motherhood should be a woman’s prison [35] (p. 26). Second, reducing women to objects aims to annul and extinguish the irrepressible margin of freedom provided by female otherness. Finally, the patriarchal ideology uses violence as a radical form of illiteracy concerning the female language of freedom and otherness. Or, as stated in Recalcati’s words, “we should think of an education to otherness” [22] (p. 29). Therefore, this model assumes the existence of inevitable interconnections between the collective and the individual levels. In fact, people grow up in a world culturally and socially structured on these premises, in addition to interacting within the relational contexts through which they are formed at various levels.

Violence against women is often justified by power differentials between the sexes, which emphasize the devaluation of all that is feminine and the justification of male supremacy and aggressiveness. The general lack of recognition afforded to women’s speech and self-determination denotes one symptom of this phenomenon within social bonds [36].

However, men have recently been increasingly involved in the active search for solutions through promoting information, education and a culture of non-violence and equal rights, in addition to taking part in treatment programs for perpetrators of IPV (Committee of Ministers to member states on the Council of Europe Action Plan, Istanbul Convention, Article 16, Points 1, 2 and 3). This trend has been recognised as a key factor, as when “preventing male violence against women, cultural changes are particularly important” [29] (p. 6). These changes not only must lead to overcoming the power of stereotypes, but also, and above all, must force men to look within themselves, to listen to and to understand their emotions, and to confront and accept their own frailty and weaknesses, ”things that women have long learned to do” [29] (p. 6).

### 2.2. The Organizational and Relational Levels

The organizational level detects actions and interventions that could reduce the impact of gender violence. Arcidiacono and Palomba [37] refer to Bronfenbrenner’s four-system model in describing the contextual level of child abuse, providing evidence that the latter does not always have a therapeutic impact on children and their pathological family system, but, on the contrary, often causes secondary damage. Furthermore, a lack of protective measures often leaves women unaccompanied when facing their violent partners. Similarly, in the case of child abuse, “attention should be paid to the overarching system constituted by society, family and institutional personnel, together with the legal bodies that deal with the effects of IPV throughout the different stages of intervention, which are, in turn, influenced by the macro-context” at an organization level [37] (p. 65). 

The organizational level expands how services, such as those provided by social services, public security officials, parishes, schools, neighbours, extended families, doctors, hospitals and the workplace interact with one other and produce resources and support for the direct or indirect management of IPV and its effects on all family members. The literature also shows the importance of receiving psychological diagnoses in an emergency department that is equipped to detect psychological violence and all potential related effects [38].

Interaction among services is essential, because it offers support, resources and participation, creating a functional network to end IPV. It is also evident that service provision is strictly related to cultural beliefs and legal rules. Furthermore, the literature focuses on the effectiveness and validity of treatment programs and their organizational framework, as well as on the quality of collaboration within this functional network [39,40,41,42]. 

At the organizational level, interconnections among services that can counter IPV in a territorial community are a very significant issue. However, support among services and collaborative participation in IPV management are often lacking for the reasons described below. The assistance of the ViDaCS project team was essential in gauging the support and participation of IPV-related services, as they generously shared the results of their research on IPV victims and perpetrators with us [43]. This research consisted of 50 interviews of staff members at centres focused on IPV prevention, management and treatment. The respondents were between the ages of 27 and 70 and consisted of both volunteers and professionals possessing between one and 45 years of experience, and who worked in a variety of different professional roles, encompassing psychologists, psychotherapists, social workers, honorary judges, technical consultants, regional councillors, family mediators, educators, lawyers, criminologists, nurses and first aid doctors. 

These interviews raised specific suggestions concerning service interaction and personnel in dealing with IPV and its effects, in addition to how they deal with the perpetrators and victims of violence.

#### Male and Female Personnel Characterization

Personnel’s feelings of fear and anger towards perpetrators, as well as their collusion with perpetrators’ “modality of denial,” could lead them to develop either an attitude of refusal towards perpetrators or, conversely, to minimize their violent acts. Therefore, working with men who have committed violence means maintaining constant awareness of this paradox, in addition to knowing how to support them by working with them to recognize their ambivalent and violent attitudes and actions, while also providing them with opportunities to change and transform themselves into better men, simultaneously safeguarding their positive self-image. 

Female staff, most frequently, work with male perpetrators of IPV using a strategy of ‘perceived listening,’ rather than empathy, which is a type of listening that allows for the maintenance of emotional distance and a more neutral attitude towards men. Women seem to not place significant importance on the gender of the staff member working with perpetrators of violence; however, men report that when staff and perpetrators have the same gender, it allows for them to better reflect on questions of masculinity and empathy, which are considered to be very important [44].

Female staff at women’s centres and shelters are characterized by an empathetic relationship with victims, who are often seen as women who were “destroyed” by men [43]. Their vision is influenced by the need to encounter an urgent solution designed to get female victims and their children to safety, away from perpetrators of IPV.

In ending violence, staff members’ gender is important; in fact, identification issues are central in taking charge of both victims and perpetrators of violence. When male staff work with male perpetrators of violence, it leads to a shared reflective awareness of the importance of gender roles. Collaboration between female staff and victims often results in feelings of rage against perpetrators, as well as against any women who collude with them. Female personnel who work with women in emergency situations on a daily basis express the need to immediately assist victims, particularly considering the low levels of economic resources reserved for women.

Sometimes, the funds spent on men’s treatments are perceived as being “diverted” from more urgent priorities, especially considering the need to protect women and take care of children who have been exposed to violence. 

It is within the relational system between couples, which is based on the control and domination of men over women, where psychological violence plays an important role, and women who experience conjugal violence live in a climate of continuous tension [45]. The intervention procedures adopted by personnel are also considered at a relational level. The perceptions and representations of personnel dealing with IPV, their experience in meeting violent partners, but also victims, and methods for providing care and related treatment have only recently been explored.

Service personnel involved in preventing gender violence consider working with men to be a major challenge [44,46,47]. Their representation of perpetrators reflects men whose behaviour possesses a cultural matrix that responds to a model of patriarchal prevarication towards women and/or a psychiatric condition [47]. Perpetrators are viewed as men who are unable to recognize their violent behaviours and who also deny them, while seeming to lack any awareness of the pain caused by both of these factors to their female partners and children. As a result of their denial, male perpetrators need access to treatment to enable them to become aware of their thoughts and actions. This means that they must be encouraged to undertake a path of change, being induced to develop awareness of their ways of thinking and acting. These are men who need access to treatment, but in a “spintaneous” way (from the word “pushed” in Italian, combined with the word “spontaneous” in English and Italian) [47]. According to this representation, people working with perpetrators of violence need to be both sensitive and competent in the management of their own experiences of violence and in assisting men to cultivate awareness and assume responsibility for their actions, while also working within a network, in cooperation with various institutions, law enforcement agencies and social services [47]. 

Furthermore, our findings emphasized the importance of the manner in which staff enter into relationships with men responsible for violence within treatment programs dedicated to them [46]. The authors highlighted the importance of personnel examining the historical, social and political aspects of their experiences, acknowledging their internal values and assumptions in a research and professional intervention context and how they could influence their professional practices [48,49]. In working with violent men, it is necessary to reflect on the sociocultural significance of violence and to evaluate relational obstacles and resources. Violence begins and grows in relational systems and can only be fought within them. Perpetrators of violence have to be involved in preventing gender violence through their participation in awareness and prevention programs, as violent acts are not solely a momentary lack of control, but they are concealed in an escalation of events that develops throughout personal and relational stories. Violent acts result from elevated feelings vulnerability, covert narcissism, and, especially, an inability to think about and tolerate frustrations [43]. 

These characteristics of men, as reported by IPV care providers, fit together with those of women victims of violence, who are represented as being highly dependent and frail, in addition to possessing low self-esteem, strong feelings of impotence and weakness and, often, highly collusive desires. This collusion, however, leads staff who work with women to feel practically helpless and very disconcerted [50]. Working with male perpetrators and female victims of IPV is further complicated by the persistence of a social representation of the traditional family system, which considers intimate partner violence to be a private issue, and; therefore, feelings of loneliness within their families make life more difficult for both the victims and actors who experience violence [29].

Based upon the aforementioned research, [44], it has emerged that male perpetrators of violence report some willingness to be involved in confrontational and sharing environments managed by male staff. This leads to the hypothesis that spaces for mutual aid and/or a ‘male’ space lead to greater efficiency, as “dialogue with women seems to be a goal, not a starting point” [51] (p. 171), and male personnel could be a resource in working with perpetrators of violence. In fact, Amodeo et al. [44] showed that male staff perceive the need to recognize themselves in the ambivalent aspects of violence more often than female staff, emphasizing the importance of contact with one’s own emotions and emotional experiences. They presented a more incisive reference to gender identity and reported that working with violent men “means recognizing the closeness due to gender belonging and, through work reflecting on oneself, positively reshaping and restoring gender identity. This also allows them to distance themselves from the image of violence stereotypically attributed to men” [33] (p. 19). As a result, male staff were more likely than women to recognize the need to develop self-reflexive professional skills that allow them to maintain a “binocular view,” oriented towards both themselves and others. With competent assistance, personnel must also learn to recognize their own inadequate feelings, ambivalence and possible insensitivity to the pain of others. However, recognizing this negativity leads to the risk of seeing their own image destroyed, which can lead to radical self-loathing and a desire for self-destruction [50].

### 2.3. The Individual Level 

Considering gender violence from an ecological perspective, our operational model will focus on its trigger factors and their effects, in addition to analysing the roles and responsibilities of everyone affected by it, not only its victims. 

The ViDaCS project report and dataset [43,44,45,46,47,48,49,50,51,52] we examined guided our focus to specific aspects related to couple members and their offspring.

*Women and IPV*. The psychodynamic approach examines the intra-psychic processes of individuals [53,54]. From the perspective of this approach, violent behaviour is an external manifestation of a pathological variance of aggression that creates a violent relationship where women victims of violence are subject to a number of emotional factors, particularly shame, fear, guilt and terror [30].

In general, there is a “silence of death” among women [55] that reflects a social and political patriarchal dimension that trivializes or euphemizes violent acts and makes women feel guilty. This silence originates in the unconscious alliances that structure the primary relationship and in the mother‒daughter bond, in particular. There, the symbolic transmission of bodily and sexual aspects is characterized by secrets and the “unsaid,” which structure taboos and feelings of guilt and shame linked to female sexuality [56]. Consequently, there is a very strong interconnection between the individual psyche and the sociocultural system in that both are structured on adherence to this silence and, therefore, to the varying forms of gender-based violence. 

Violence against women has a ‘cyclicity’ that manifests itself through three phases: the development of tension, explosion, and forgiveness leading to reconciliation. It is this final phase that keeps women tied to men without changing the modalities of their relationships. Indeed, it is forgiveness that preserves connections between women and their partners following violent acts, perpetuating violence. Following completion, this cycle will begin again and moments of tranquillity will become shorter and shorter over time [57,58,59]. The continual exposure to this cycle of violence at the physical, sexual and/or psychological levels often leads women to develop Battered Woman Syndrome, a symptomatology identified through six criteria, which include intrusive memories of traumatic events, high levels of anxiety, avoidance behaviour, the interruption of personal relationships, a distorted body image and problems within the intimate and sexual spheres [57,58].

Women report their initial feelings of fusion with their male partners and of being in an idyllic relationship that they characterize as completely satisfying, describing their male partners as a sort of ‘Prince Charming,’ who only later becomes a different person. Violence creeps in slowly and is often justified by men and tolerated by women, in the name of their initial feelings of love, which are difficult to abandon. This attachment is supported by women’s unhelpful idea that only they can save their male partners [29]. This situation encompasses emotions of tolerance, effort and endurance, but also a silence, which, in its unconscious and political dimensions, traps women in a taboo that becomes symbolic of violence [55,56,57,58,59,60] and strengthens the context of male domination [61,62].

This same cycle of violence ”can often annihilate and deprive subjects of their words and thoughts” [60] (p. 113). It is not possible to identify a particular moment or gesture that will motivate women to act, but there is an individual limit of tolerance beyond which women decide to leave a relationship and turn to various services for help [29].

However, women seek the assistance of social services only when their survival instincts begin to prevail, which often occurs when men’s gestures reach an extreme. This can provoke women’s instincts to protect their children and lead them to react. In this situation, fear sometimes turns into anger, and women often decide to save themselves and their children rather than their male partners.

*Children and adolescents*. Many children who grow up in a violent context become very suspicious of others and silent, while others tend to express their anger through challenging and provocative behaviour. 

As stated by the ViDaCS project [43], staff working with children who have witnessed IPV report that children often feel like “little adults” who have to “take care” of their parents and be careful not to feed conflicts [52]. At home they learn to understand how to prevent conflict and to modulate their behaviour and that of their parents to avoid quarrels. Additionally, they are always attentive and often feel a great deal of tension. During violent episodes, some children tend to hide, run away to another room, or plug their ears, so as not to hear, while others seek the protection of one of their parents (especially their mothers) in an effort to stop the violence.

Children become “invisible” during moments of conflict between their parents, as the latter stop “seeing” children during quarrels because they are too absorbed by a conflict that overshadows their vision and estranges them from reality. Violence always involves psychological upheavals, as well as the same premises of prevarication and possession that lead to violence against women.

Mothers often tell social services about children who no longer sleep well, do not want to go to school, do not want to be away from them, or, sometimes, ask to stay at relatives’ houses and do not wish to play. Very often, teachers are the first to notice a change in children, and to report it to their families or to social services, often causing mothers to feel shame at this point. Child witnesses experience psychosocial, emotional and cognitive effects, in terms of memory and learning, and; therefore, they (re)produce negative behaviours. Indeed, they are more at risk of internalizing certain disorders (withdrawal, anxiety and depression), while externalizing other disorders (delinquency and the perpetration of violence), both of which should be viewed as maladaptive responses resulting from the way in which individuals interpret violence [15]. Respectively, they can lead to re-victimization or the perpetration of IPV in adolescence and/or as an adult [63].

Various emotions are, commonly, expressed by children through drawings and stories related to their parents’ quarrels [52]. Many children evoke a fear of separation, encapsulated by the loss of their mother (through death or removal) and, sometimes, of their father, or as a result of state-mandated removal from both parents. Child witnesses often show anger towards their father, the perpetrator of violence against their mother, who, in turn, “feeds” the arguments by responding to provocations and bringing them to facilities or anti-violence centres. In other instances, children demonstrate antagonistic feelings towards one or both parents because their abilities of protection and restraint have failed. Furthermore, guilt, a sense of impotence, despair and disorientation are all emotions commonly observed in child witnesses of IPV. 

In this regard, witnessing violence is a specific form of violence where children are indirect targets. It is worth mentioning that the aim of the ViDaCS project [52] is to raise awareness of children’s discomfort and pain in a way that may lead fathers to seek out assistance and/or to partake in psychotherapy.

*Men*. Men are often represented by fragility, pathological narcissism and difficulties in communicating and reasoning. These are men who act out their emotions, as they find it difficult to tolerate the reasoning process. This confirms what the literature defines as a lack of emotional regulation, self-regulation [64,65] and interpersonal skills [66]. Often these men have been witnesses or victims of violence since childhood. Therefore, a special focus should be given to devising how to intervene before trigger factors for the escalation of violence occur (see https://www.facebook.com/vidacsEU/) and, furthermore, how to help perpetrators in dealing with their violent behaviour towards women.

Mizen states that “violence represents not an absence of mind, but rather an ablation of mind where an individual struggles in the face of affective experiences that they are unable to manage, in addition to being unable to divest themselves of important qualities of mind through psychological defence” [67] (p. 417). Therefore, violent behaviour can be characterized as “the subjective psychological and especially affective substrates of violence that are brought to the fore” [68] (p. 1). A sense of self-annihilation and impotence driving intolerable and unbearable fears, such as the experience of pervasive panic, form the backdrop for violent acts. When deprived of their primary defences, individuals are unable to reflect on (mentalize) their thoughts. In men, it is absolutely necessary to disempower and weaken perceived internal threats of annihilation closely linked to murderous/suicidal fantasies and to activate an emotional self-regulation process [69]. 

In fact, violence is an expression of the failure of the psychic process of integration, which concerns the integration of emotional experiences. It leads men to project the divided and hostile parts of themselves, which they can neither think about, nor elaborate on, onto their partners through a process of projective identification, thus allowing for the expulsion of their intolerable internal parts. Among violent adults, this mechanism is used as a defence in the absence of an ability to reflect upon their emotions. From this perspective, the other becomes a depository for , these parts of violent adults, so that any change, departure, or non-response threatens their sense of self and, potentially, can cause the disintegration of the self. According to Mizen, the same apparent strength of these men is provided by the construction of an “exoskeleton-Self,” a structure built on the weakness and fragility of an “endoskeleton-Self,” which results from the lack of an ability to tolerate, give meaning to and mentalize frustration. This mechanism creates a hostile context around men, but, at the same time, by involving parts of the self, it keeps men inevitably bound to their female partners [70,71].

Similarly, in their research into 35 families that experienced abuse and 35 families that did not, Blair Justice and Rita Justice [71] found that people who hailed from ”abusing families” had experienced family relationships characterized by a lack of ”emotional” communication and support, as well as life crises due to some features of their parents, such as an inability to adjust and control emotions during their childhood. In these cases, these experiences lead people to continuously search for care and support, due to their strong need for love. They can also cause desires to exercise hyper-control over others (partners and/or children) who are called on to compensate for these shortcomings. Undifferentiated personalities looking for symbiotic relationships are the most frequent consequences of these stories and their results, and can sometimes allow for repeated instances of child abuse. Similarly, in witnessing violence, children become an unexpected target of violence among couples. Therefore, children are an invisible and unforeseen witness of emotional troubles. 

Concerning perpetrators’ possibilities for change, their potential motivations are strictly attributed to the presence of children, similar to cases where women decide to leave abusive relationships. However, fatherhood is seen as a “double-edged sword” and as a grey area, as it can also become a cause for violence, often commencing with pregnancy [72], the object of emotional blackmail and a specific instrument to exert control over women. The threat of losing their female partners and/or their role as a father, the loss of a loved one (e.g., a parent) and internal and authentic motivation (supported by insight) are considered to be further valid motivations to continue their treatment and to pursue change. Extrinsic motivations, such as those furnished by law enforcement and court orders, or by a partner’s wishes, increase the chances of either not participating in treatment or non-compliance, in the short term.

According to the literature [34], social support in these cases can be considered a promoter of change among these men. In fact, the maintenance of positive relationships outside the intimate violent relationship fosters positive change. In cases where perpetrators explain their situation to family or friends, their motivations to deal with their violent acts are stronger and they are more likely to change [73]. Moreover, the literature shows that when men’s relationships with their female partners are undermined and they no longer feel that their partners value them, their attitude of blaming their female partners and denigrating their connections with them keeps them from seeking treatment [74,75].

Many men claim to have chosen the wrong partner, one who brings out the worst in them, stating that, otherwise, they would never have resorted to violence [76]. Men feel entitled to abuse and control, and they often fail to recognize the impact and severity of their behaviour on their partners and children [75]. Perpetrators’ difficulties in adhering to treatment programs have led researchers and professionals to focus on motivational factors, which, however, always require men to recognize their violent acts and to assume responsibility for them. Providing motivation for change is particularly complicated in the treatment of violent perpetrators, due to the fact that even many staff working in the sector, especially those who work exclusively with women, possess highly sceptical and resigned perceptions of this strategy.

## 3. Operative Ecological Reciprocal Interactions

As described in the introduction, the specificity of our model is its ability to define recommendations on the organizational level, which we consider to be the core level for activating procedures to end IPV. We were able to identify several areas and strategies lacking resources from the results obtained by the ViDaCS research related to the organizational level. Among these, the absence of effective communication between the services that deal with victims and those that deal with perpetrators stood out. Furthermore, among the respondents working in children’s services and women’s shelters examined by this research [43], a pattern of poor awareness of current interventions and treatment for perpetrators of IPV emerged. This could be due to either the specificity of the work setting of the respondents (e.g., emergency anti-violence centres), or to an individual lack of interest and awareness concerning the issue. All respondents highlighted the importance of and the need for sharing and collaboration among the various institutions and organizations active in the field, even though there appears to be a lack of communication between anti-violence centres for women and children and those dedicated to men. Additionally, our research highlighted the detrimental impacts of distrust towards the possibility of effective collaboration among different services. Resignation towards the difficulties faced in enacting change in men who perpetrate IPV highlights the conundrum posed by the already scarce funds dedicated to anti-violence centres for women and the risk that projects aimed at perpetrators will be funded from this same general budget. Moreover, our research underscored the absence of a common and integrated model to understand IPV. From the qualitative data collected [43], it emerges that people working at gender violence prevention centres do not share a common model for understanding violence. Personnel working exclusively with women and/or children possess an emergency perspective that is focused on removing women and children from violent situations and from their roles as victims, allowing them to be properly protected and supported while punishing their persecutors. In contrast, centres that work with men tend to be more concentrated on preventing future violence. The need for emergency action to protect women and children makes communication among different services that combat gender violence even more difficult. 

In contrast, having a common model and an interwoven strategy would facilitate the creation of a network that includes services to prevent gender violence by intercepting invisible perpetrators and raising awareness of these services, in addition to the ways in which they can request help.

## 4. Conclusions 

According to the practice and theories of community psychology [77,78,79], Kelly’s ecological approach focuses on the intertwinement of four dimensions comprised of interdependence, the cycling of resources, adaptation and succession at each level. Below, Figure 1 provides evidence of some cyclically repeated factors and of the interdependence among the different dimensions and levels. Thus, it is evident that changes in one component reverberate throughout the entire system. Individual factors cannot be considered independently from organizational and relational issues, nor from the Zeitgeist. They are also capable of affecting an individual’s symbolic representation, cognitions and emotions, as one’s social environment affects the relational issues in an individual’s life, including how resources are defined, created, distributed, used and transformed. In this regard, gender violence is rooted in the social norms that define women and men’s expected behaviours, as women and men’s reciprocal interactions are imbued in social norms, and individual behaviours are strictly related to more hidden and unconscious beliefs and social expectations. Therefore, gender violence could be considered as a maladaptation that necessitates coping, adaption and changes in the relational environment, which cause men to become perpetrators of IPV and women to become victims.

Furthermore, according to the latest dossier published by Save the Children, 427,000 minors in Italy have experienced violence and/or exposure to it at home in the last five years. This is a disconcerting fact when one reflects upon the primary functions of protection, care and growth that are attributed to the family.

Traditionally, the literature has focused on the consequences of the direct abuse of minors, neglecting child witnesses of violence, as the latter are much more difficult to recognize, research and evaluate through statistical surveys. This difficulty is also supported by the socially and culturally shared conviction that children “do not feel” violence. Parents underestimate the gravity of the experiences and consequences that affect children who witness violence, given that children’s awareness of abuse is much greater than parents and society acknowledge. 

Children are neither passive nor indifferent to violence [15,80,81]; although sometimes they do not witness it directly, they are fully aware of its occurrence [15,81,82]. Living in a violent family has negative implications for children’s mental and physical health, both in the short and the long term [15,83,84]. This is primarily because adults can become violent even against children, but also because parents are unable to satisfy their physical, emotional and caregiving needs; in addition, children could be physically injured during a quarrel between parents, or could become an object of contention between them.

Children are perceived by professionals as being ‘overloads’ on their parents’ experiences, especially those of their primary caregiver, who is most often a mother in need of protection, care and support [52].

This ecological model assumes that individual well-being can only be reached if the relational, organizational and collective levels are integrated. 

Figure 1 summarizes the interventions and goals at different levels, showing reciprocal interactions. It describes an integrated preventive strategy that highlights the importance of furnishing protection to women, while simultaneously reaching invisible, violent men and, therefore, being able to enter into contact with their emotional world. As a result, preventative strategies emphasize that others should not be held accountable for the violent behaviours of perpetrators. Actors of violence must be sanctioned and punished, but, at some point, it will be necessary to consider “truth and reconciliation” interventions between men and women, perpetrators and victims. The experiences of similar interventions in Uganda, South Africa and Morocco that were designed to reconcile populations that had experienced years of violent abuse inflicted by one population on another may offer some ideas for dealing with devastating family situations affected by violent behaviour. 

In light of this assumption, the following recommendations for the treatment of perpetrators and victims take into account the four levels considered in the theoretical‒operational model described above. 

### 4.1. Guidelines to Support the Development of Perpetrator Treatment and Professional Capacity Building 

The monitoring of the effectiveness of services and the main recommendations to end IPV and its effects are detailed below for each level.

#### 4.1.1. Collective/Community Level

At the collective level, we suggest interventions to promote a new cultural system based on the values of respect and fairness between men and women, with the goal of ending gender violence and IPV, in particular. The following three specific areas for action are suggested.■Policies
a)Develop policies that improve upon the actions of the legal system and enhance cultural awareness to promote social change;b)Train educational, health and social services professionals to deal with their own fears related to acknowledging the existence of violence and to overcome their preference to neither see nor hear its effects.■Educational interventions
a)Formalize a protocol outlining modes of collaboration with educational institutions to disseminate the values of respect and fairness with the aim of deconstructing traditional gender roles rooted in the patriarchal culture.b)Support laboratories using participatory and experiential methods that are aimed at emotional experiences and which will facilitate access to the most intimate social representations learned in family contexts.c)Enhance interventions at schools intended to create a network between schools and the services dedicated to uncovering cases of IPV, in order to prevent its effects on children’s well-being and social integration.■Promote family well-being

In this regard, we recommend promoting a culture of family wellbeing where mothers are respected and fathers’ responsibilities are more visible and recognized in politics, history and practice [85,86]. At a collective level, this signifies a recognition of their paternal role in promoting family well-being, in addition to increasing men’s awareness of their own responsibilities [85,86].

#### 4.1.2. The Organizational Level

At the organizational level, our recommendations aim to:a)Promote a collaborative network between the services that deal with gender violence, victims and perpetrators;b)Define common and synergistic procedures in order to avoid the fragmentation of interventions against violence, especially when the financial resources allocated towards this goal are limited;c)Support the development of perpetrator treatment and professional capacity building, and;d)Sign a memorandum of understanding with local police, urging stronger collaboration among the network that precedes victim intake and introduction to intervention services, directed at perpetrators who have been subject to legal action.

Furthermore, our research highlights the need to:a)Increase health and legal services and projects, where personnel are able to recognise the severity of perpetrators’ treatment and to use evaluation tools to define their dangerousness, such as the SARA (Spousal Assault Risk Assessment) approach, outlined by Baldry [87]. Furthermore, health professionals and law enforcement officers should be able to recognise the severity and potential repetitiveness of IPV using the appropriate tools, such as VITA (Intimate Violence and Traumatic Affects Scale [30]), a self-reporting instrument used to assess the intensity of the post-traumatic affects deriving from IPV.b)Create reflection groups in service contexts to support personnel dealing with gender violence.c)Enhance the effectiveness of protective measures provided by services directed towards women, particularly among:i)Emergency units able to recognize signs of physical and psychological violence;ii)Effective opportunities for protection (i.e., shelter houses);iii)Opportunities for economic independence (job placements, etc.).d)Support services focused on men to discover “invisible perpetrators,” who may be the bedrock of femicide.

Moreover, concerning women, there is a need to deal with the denial of their own suffering, shame, guilt and terror. 

At a relational level, these recommendations are directed towards staff and, in particular, towards everyone who has contact with perpetrators of violence. Specific training will be necessary, and training programs should strive to:a)Establish a specific model for understanding violent behaviour;b)Increase reflexivity, allowing personnel to become aware of their own emotions and thoughts, while facilitating their ability to avoid taking a one-sided position;c)Support staff in coping with the denial mechanisms typically used by perpetrators, while encouraging perpetrators to reduce their denial mechanisms, increasing their awareness of the harms caused by witnessing violence; andd)Increase the staff’s skills in facilitating processes designed to make perpetrators aware of their violent actions and to help them find new strategies for coping with negative emotions.

We also suggest the introduction of discussion groups among staff, in order to provide them with a time and place for group reflection on their experiences in treating perpetrators. Furthermore, there is the need to draw greater attention to the prejudices and perceptions of the perpetrators of violence, often shared by the staff, in addition to the denial and disdain that men often bring into treatment.

Specific recommendations are aimed at reaching the “invisible” perpetrators, focused on those who are not involved in the legal system. Invisible perpetrators are those who are socially well-integrated, but who cannot refrain from violence in their intimate relationships. Our guidelines suggest that interventions should augment perpetrators’ awareness of what happens “a moment before” their violent actions, in addition to increasing their motivation to request help through emotional experiences that stimulate their paternal responsibilities. Finally, the end of the patriarchy poses new challenges for men and women, creating the need to develop new forms of reciprocal relationships to overcome each new risk, fear and addiction.

Figure 2 summarizes the different steps of our ecological model, highlighting the goals, actions and actors and their interdependence within the given time frame. 

Last, but not least, services aimed at protecting victims and those designed to prevent and treat the violent actions of male partners must interact when handling these cases. This is the first step towards a new pact between men and women designed to prevent gender-based violence and its effects on emotional and relational life. 

## Figures and Tables

**Figure 1 ijerph-16-01652-f001:**
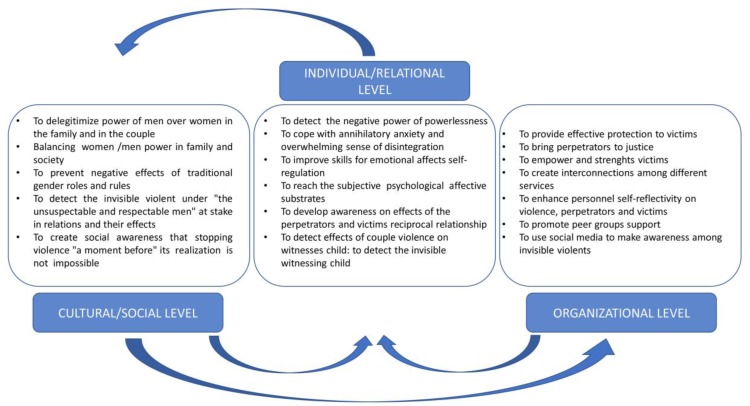
The ecological model and operationalization of goals and actions at different levels.

**Figure 2 ijerph-16-01652-f002:**
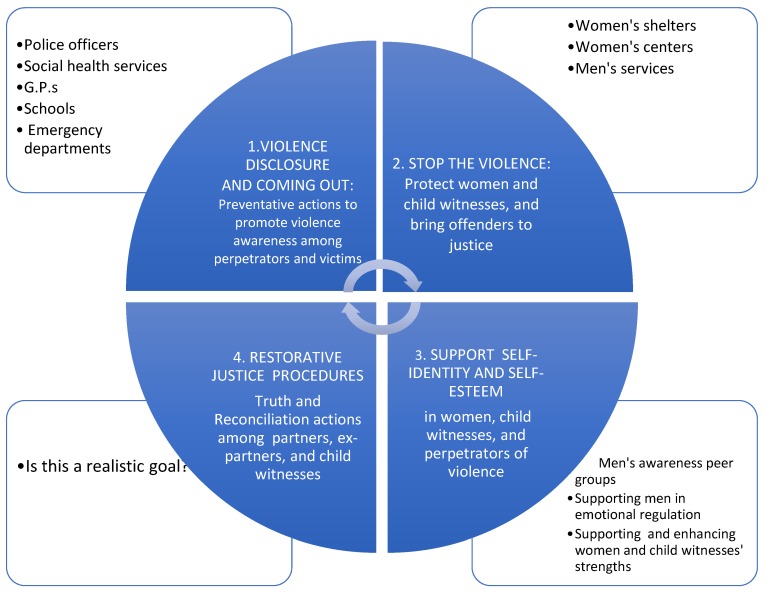
The ecological approach: goals, actions and actors at different levels in terms of their interdependence and within the model time frame.

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
