# Peer review of "Ending Intimate Partner Violence (IPV) and Locating Men at Stake: An Ecological Approach"

_ijerph, 2019, doi:10.3390/ijerph16091652_

Round 1
Reviewer 1 Report
The manuscript “An ecological approach to gender Intimate partner violence (IPV): perpetrators at stake” analyzes a topic of interest and current importance, such as the IPV.
This article provides a theoretical proposal based on the ecological model. Considering the ecological model, authors integrates the following theoretical approaches: social approach, feminist approach, cognitive-behavioral approach and psychodynamic approach and the implication of their joint impact. However, there are important questions that should be addressed:
The most important question is that the rational of the theoretical proposal is quite confusing and needs to be explained more in depth. The idea should be better articulated. The contribution of the manuscript is not clear. Why it is innovative and interesting to integrate the proposals mentioned above into the ecological model to explain IPV.
On the other hand, authors propose recommendations to support the development of perpetrator treatment and professional capacity building actions. However, it is not justified, and it does not seem to be one of the objectives of the paper.
ANOTHER ISSUES:
- Headings and subheadings are not clear
- There is a mix of concepts that are unexplained such as: Gender intimate partner violence, Intimate partner violence, and Domestic violence.
- Authors should take in consideration the template and the guide for authors.
Author Response
1) We improved the rationale. Your comments let us understand that there were many implicit concepts that prevented a clear understanding of the text. We hope that now it is better.
Line 67-- 108. we add:
1.1 IPV and Gender- based violence
Our ecological model will set out the reciprocal interconnections of the different variables acting at specific levels.
The innovative dimension of the article is then to frame in an ecological perspective some specific indications for social and psychological intervention. Therefore this article will specifically operationalize the ecological principles into specific operational guidelines for the organization of services.
The gender-based perspective that we assume stresses that men and women don’t share the same power in the couple and in the family at large. It is a fact that couple conflict degenerates into violent actions against women which sometimes leads them to being killed. Even the recently introduced term feminicide has no equivalent for the masculine.It is no coincidence that violence can be elicited or aggravated when the woman assumes subjective roles that depart from these fixed and gendered role assignments. In fact, in these cases the perpetrators of the violence voices fantasies of the cruel mother or narcissistic woman. Moreover, women fight against male supremacy or the challenge to its authority is experienced by men as a primary offense that make them intimately vulnerable and impotent canceling their deepest identity. Therefore, the advent of more balanced social power makes men unable to cope with a new society where male privilege is broken down.
The Figures on statistics gives evidence of the distribution of the phenomenon in the population and its prevalence among male partners [ 7 ] but if we focus only on psychological dimensions of perpetrators, the issue could be understood as only an individual or relational pathology; therefore we need an ecological approach which is able to inscribe a more individual dimension in a more general social power issue.
On these grounds, this integrated and holistic theoretical and operational model for understanding and intervening in IPV will also overcome the fragmentation of services directed at IPV due to their different goals. This ecological approach indeed proposes a perspective that places value on interventions directed not only at victims but also at perpetrators, both in preventative and therapeutic perspectives.
Moreover let's introduce on the couple scene the witnessing child. Since the 1970s the phenomenon of witnessing domestic violence was considered within psychological mistreatment (McGee & Wolfe, 1991; Peled & Davis, 1995; Somer & Braunstein, 1999), because it significantly disrupted child socialization (XXXMeta-Analysis article) and brought about emotional, psychological, behavioral and social problems (e.g., Fantuzzo & Lindquist, 1989; Jaffe, Wolfe, & Wilson, 1990; Kolbo, Blakely, & Engleman, 1996; Margolin & Gordis, 2000; Wolak & Finkelhor, 1998). . Even if not all children exposed to witnessing violence developed problems, often they developed a conflict resolution method characterized by the use of violence and very often they believe that they are themselves responsible for the violence of parents (Jaffe, Hurley, & Wolfe, 1990). The consequences of continuous exposure to inter-parental violence also differ according to sex and age: girls would be more at risk in adolescence, while males would be more at risk during childhood. Children witnessing violence also express behavioural and relational difficulties, therefore they have recently been recognized as a specific target for treatment and preventative interventions.
See in the text all further modifications and most important shifts in revision form.
2) see line 70-73 We add: The innovative dimension of the article is to frame in an ecological perspective some specific indications for social and psychological intervention. Therefore, the article is operationalizing the ecological principle in the field of IPV
See in the text all modifications and shifts in revision form
3) We clarify the goals of the article in the introduction and in the abstract emphasizing the need for recommendations to support the perpetrators and increase professional capacity building
see in the text all modifications, shifts and the redefinition of the abstract
AN OTHER ISSUE
We revised Headings and subheadings according to the Journal rules.
We clarify the use of gender-based intimate partner violence in the sense of giving value to the fact that intimate partner violence is strictly related to social role representation and attribution in a given society, however to be more clear we talk about gender based violence as stated in the Istanbul document. We used intimate partner violence to differentiate from violence committed by men without any specific relation to the victim; we specify Intimate partner violence within domestic violence introducing children as witnessing IPV. Your comments let us identify that our definition led to some misunderstanding. Therefore, we amended the text and define IPV within domestic violence, and we make it clear that we are also considering its effects on witnessing children.
5)We took into better consideration the template and the guide for authors.
Reviewer 2 Report
Thank you for the opportunity to review this paper. This is an excellent conceptual piece of work that reviews findings on a significant and pertinent topic.
Author Response
The reviewer said that fine/minor spell check is required. We have had this done an English mother tongue proofreader.
Reviewer 3 Report
This article is for the most part well organised and reasonably well written.
However whilst I appreciate this is a conceptual paper I am uncertain the extent to which this is novel/new, or makes a contribution to the literature.
Some points of concern.
1) What are the limitations of the current theoretical models that you discuss, and perhaps those that you don’t incorporate into your ecological model? Am not convinced of why the model you propose has additional benefits – I suggest you need to explicitly draw out its utility and novelty compared to other approaches.
2) This model is focussed on traditional male on female violence. This should be specified at the outset. I would also suggest it requires a section explaining why this only applies to this type of violence (IPV), and for example is not appropriate for female on male IPV – which is less recorded and less discussed in the literature – or how can this be adapted to fit this.
3) (Lines 56-79) I would avoid use of term ‘fight’ domestic violence. Prevent/reduce might be better alternatives.
4) Line 127 – Gender violence is then inscribed in the racist matrix – why racist? I do not follow. Do you mean hate crime – I am not sure where race comes into this discussion
5) Lines 143-146. What is relevance of this last paragraph? Please spell out the point you are trying to support
6) In section 6.2.2 you being to discuss child abuse. Is this not a very different form of IPV to adult to adult. I found this rather confusing.
7) Line 180 – what research are you referring to here?
8) Line 192 – what qualitative data are you referring to
9) Recommendations section 3. I appreciate that for each element of your model you have tried to identify prevention lessons. However, some of this discussion felt rather thin and could be better thought out. Moreover, can these be taken in isolation. How does this model handle complex interactions between the community, the organisation, and the individual
Author Response
This article is for the most part well organised and reasonably well written.
However whilst I appreciate this is a conceptual paper I am uncertain the extent to which this is novel/new, or makes a contribution to the literature.
Some points of concern.
1) What are the limitations of the current theoretical models that you discuss, and perhaps those that you don’t incorporate into your ecological model? Am not convinced of why the model you propose has additional benefits – I suggest you need to explicitly draw out its utility and novelty compared to other approaches.
Our model gives indications at organizational level to improve interactions and connections among different services. According to the Istanbul document the European council very clearly expresses the need to protect the woman and persecute the perpetrator. Therefore, its application in all countries led to the opening of women’s homes and, shelters and antiviolence centers. The need to persecute the perpetrators recognizing his act as an offense against the person hindering human rights did not focus on preventative measures for men. However, there is now an increased awareness about the need to introduce preventative measures and support. Also specific treatment for child witnesses to violence, even if this was already considered as mistreatment, has been taken into consideration only recently.Therefore, our final aim is to let services interact and create preventative interventions able to reach violence “the moment before” its realization. According to the Vidacs team our model proposes an effort to increase competence in intervening just “a moment before”.
1) This model is focussed on traditional male on female violence. This should be specified at the outset. I would also suggest it requires a section explaining why this only applies to this type of violence (IPV), and for example is not appropriate for female on male IPV – which is less recorded and less discussed in the literature – or how can this be adapted to fit this.
In this last version we made it explicit that we refer mainly to violence of men explaining the reasons for this decision and we introduced a specific paragraph on IPV and gender based violence.
2) (Lines 56-79) I would avoid use of term ‘fight’ domestic violence. Prevent/reduce might be better alternatives.
We use the term ending to include both, as well as prevent
3) Lines 143-146. What is relevance of this last paragraph? Please spell out the point you are trying to support
In this version we changed some words. “The concept is that, violence against women is legitimized within adherence to traditional gender norms and roles. Specifically, the woman's adherence to the female gender role allows her to support with high tolerance physical, sexual, psychological and emotional violence [15]”.
4) Line 127 – Gender violence is then inscribed in the racist matrix – why racist? I do not follow. Do you mean hate crime – I am not sure where race comes into this discussion.
We decided to delete this reference. We realize that it caused confusion and did not clarify the concept.
6) In section 6.2.2 you being to discuss child abuse. Is this not a very different form of IPV to adult to adult. I found this rather confusing.No we are not discussing child abuse. We use this case as an example. In fact, the case involved cultural vision, the judiciary system and service organizations taking care of a problem, people involved in that issue benefitting from a different treatment, that is the case of abused children in Italy following new rules of the last years. Therefore the example showed that when law and services took care of children, for the latter the abuse he /she was victim of had been overcame. He/she was supported and taken into care.
7) Line 180 – what research are you referring to here?
We add now here the reference in the text.
8) Line 192 – what qualitative data are you referring to
We added information on the qualitative data we referred to
9) Recommendations section 3. I appreciate that for each element of your model you have tried to identify prevention lessons. However, some of this discussion felt rather thin and could be better thought out. Moreover, can these be taken in isolation. How does this model handle complex interactions between the community, the organisation, and the individual
We now clarify how the interaction among the different levels works. To create a social and organizational answer a first goal is to make the “invisible” IPV visible in society, let it be recognized in the couple and by the services, in order to be able to take care of it with different interacting measures. see also Figure 1 and 2 that we introduced.
Reviewer 4 Report
This is an interesting topic for researchers who study intimate partner violence. However, some issues related to the current draft need to be addressed.
1) the manuscript does not have a well-defined purpose. I recommend clarifying the purpose both in the abstract and at the end of the introduction, as well as changing the title to adapt it better to the manuscript. If the study is aimed at describing qualitative findings, the manuscript should be structured making it clearer, interleaving the contributions with the text.
2) the manuscript would also improve if the authors clary at the beginning in the text what gap their contribution may fill. The ecological perspective is well known, hence it should be clear what is the contribution of this work.
3) the authors need to be precise when it comes to using the citations and some terms. For instance, they refer that WHO uses the term “domestic violence” to indicate that it “is predominantly between intimate partners”. However, they support this idea with a citation that describes sexual violence, both within intimate relationships and by strangers or acquaintances. This makes the citation not appropriate to support what they say about domestic violence. In addition, the term “domestic violence” includes other forms of violence such as child abuse.
Line 27, it is true that “violence by men against women is most common in family relationships”. However, the term “family relationships” include other relatives such as fathers, brothers, etc. Therefore, it is more precise to speak about intimate partner relationships.
4) Overall, the manuscript would need a somewhat broader review of the literature that have addressed this problem.
Lines 29-37, the quotation of WHO does not coincide with the citation. This quotation comes from another report.
Throughout the text, the terms “gender violence”, “domestic violence”, and “intimate partner violence” are apparently used with the same meaning. However, the authors need to delimit these terms better and decide which is best suited for their proposal. For example, gender violence is commonly used to empathize a feminist perspective, not an ecological one.
Line 180, a sentence states: “Among the respondents of this research, a pattern emerged of poor awareness of current 180 interventions and treatment of the perpetrators of IPV towards partners or ex-partners”. What research do they refer to?
Author Response
This is an interesting topic for researchers who study intimate partner violence. However, some issues related to the current draft need to be addressed.
POINT 1the manuscript does not have a well-defined purpose. I recommend clarifying the purpose both in the abstract and at the end of the introduction, as well as changing the title to adapt it better to the manuscript. If the study is aimed at describing qualitative findings, the manuscript should be structured making it clearer, interleaving the contributions with the text.
1.1 ) We revised the aims in the abstract and in the introduction. We rewrote the abstract. We changed the title as following:Ending intimate partner violence (IPV) locating men at stake. An ecological approach.
1.2)New Abstract to insert
Intervention for ending intimate partner violence (IPV) has usually not provided integrated approches. Judiciary and social policies have the urgency to protect, assist and empower women and to bring the offender to justice. Men have been mainly considered in their role of perpetrators to be subjected to judicial measures and child witnesses of violence have not been considered a direct target for services. There is now the need for an integrated and holistic theoretical and operational model of understanding IPV as gender-based violence, and for intervening to end the fragmentation of existing measures. The EU project “ViDaCs (Violent Dads in children s’ shoes)” which worked for the reconstruction and deconstruction of the effects of violence on child witnesses has given us the opportunity to collect the voices of personnel and children actors on the violence scene. Therefore the article describes implications directed at focalizing measures to deal with IPV, proposing functional connections among different services and specific preventative initiatives. Intimate partner violence will be specifically highlighted considering interventions at individual, relational, organizational and societal cultural levels.
The final goal will be to present a short guideline that takes into account the four levels considered by operationalizing the abovementioned ecological principles.
1.3 ) We explained to the reviewer that we do not repeat quotations of the qualitative research we quoted because all the research is already detailed in a cited article [30] . We gave however some more information concerning their results.
Point 2 The manuscript would also improve if the authors clary at the beginning in the text what gap their contribution may fill. The ecological perspective is well known, hence it should be clear what is the contribution of this work.
2) We explained in the introduction what our contribution may provide and what it worked for; we restructured the text and hopefully its internal strength. We prepared Figure 1 and Figure 2 to improve the readability of the text.
In the current time where femicide are increased and where a substantive equality among women and men risk to be threatened by a conservative wind, it is important to describe the intertwining of individual and cultural factors.
See Introduction: The ecological model is a general reference for intervention on gender violence (see who [1]). However, this is a general frame without specific references to the articulation of the phenomenon, all its impacts and of the service organization. Usually each model highlights some results and there is the risk of very partial visions. If we focus on psychological dimensions of perpetrators, all the issue could be understood as an individual pathology; therefore the description of perpetrators and the joint figure on statistics show the distribution of the phenomenon in the population. Some authors give evidence of the fact that also women are violence perpetrators, however an epidemiological perspective shows that only men are murderers of their partners. The need for our ecological model is to set out the reciprocal interconnections of all the variables.
Point 3 The authors need to be precise when it comes to using the citations and some terms. For instance, they refer that WHO uses the term “domestic violence” to indicate that it “is predominantly between intimate partners”. However, they support this idea with a citation that describes sexual violence, both within intimate relationships and by strangers or acquaintances. This makes the citation not appropriate to support what they say about domestic violence. In addition, the term “domestic violence” includes other forms of violence such as child abuse.
3.1)We amended the quotation line 29-37. The correct one is: World Health Organization & Pan American Health Organization. Understanding and addressing violence against women: intimate partner violence.World Health Organization; 2012. See the reference in the text. . Thanks to the reviewer who caught our wrong reference.
3.2) Line 27, it is true that “violence by men against women is most common in family relationships”. However, the term “family relationships” include other relatives such as fathers, brothers, etc. Therefore, it is more precise to speak about intimate partner relationships.
We specify what we intended with IPV and give more details about our references.
We clarify in the text the use of gender-based intimate partner violencein the sense of giving value to the fact that intimate partner violence is strictly related to social role representation and attribution in a given society; we used intimate partner violence to differentiate from violence committed by men without any specific relation to the victim; we sometimes used domestic violence to introduce and connect offspring assisted violence to the ipv. However, your comments let us identify that our definition brought on some misunderstanding. Therefore, we amended the text and use always IPV instead of domestic violence, but we make it clear that we were also considering its effects on witnessing children.
Point 4. Overall, the manuscript would need a somewhat broader review of the literature that have addressed this problem.
4) We added now and we hope that it twill be adequate.
Point 4.1). Throughout
the text, the terms “gender violence”, “domestic violence”, and
“intimate partner violence” are apparently used with the same meaning.
However, the authors need to delimit these terms better and decide which
is best suited for their proposal. For example, gender violence is
commonly used to empathize a feminist perspective, not an ecological
one.
4.1)In the introduction we clarify the terms and our use of them.
Point 4.2). Line 180, a sentence states: “Among the respondents of this research, a pattern emerged of poor awareness of current 180 interventions and treatment of the perpetrators of IPV towards partners or ex-partners”. What research do they refer to?
4.2 )Line 180 .we added now the reference in the text.
Round 2
Reviewer 3 Report
Many thanks for the changes you have made and detailed comments.
I have no additional major concerns.
A final check of English would be useful prior to submission.
Author Response
As you can see on the annex version we did again a final check of the English . To show all our English editing we add a provisional internal draft still within all tracks. Many thanks for all your support and recommendation

Reviewer 4 Report
I found the revised manuscript to be much clearer than the previous version. They have explained what advantages the ecological model offers to seek integral solutions to IPV. In addition, they have used the violence-related terms more precisely. However, the authors still need to revise the wording throughout the text so as not to divert the reader's attention from the contribution they make. Overall, they still need simplify some sentences that are unnecessarily convoluted. You can see two examples below:
Line 26: “Therefore the article describes implications directed at focalizing measures to deal with IPV,...” Did the authors want to say that the article describes measures to deal with IPV?
Lines 141-143: “The first goal is in fact to let the invisible actor or rather the socially integrated ‘unsuspected’ man, take care of his disruptive emotional experiences that he struggles to tolerate, defensively blaming his partner for his own violent behavior.”
Author Response
Dear Reviewer we did a final English check and carefully revised the phrasing.
Here enclosed a draft version with all tracks easily readable
Best regards
Caterina Arcidiacono
